# Multi-Functional Silver Nanoparticles for High-Throughput Endospore Sensing

**DOI:** 10.3390/bios12020068

**Published:** 2022-01-25

**Authors:** Shinya Ikeno, Takahiro Maekawa, Noriyasu Hara

**Affiliations:** Department of Biological Functions Engineering, Graduate School of Life Science and Systems Engineering, Kyushu Institute of Technology, 2-4 Hibikino, Wakamatsu-ku, Kitakyushu 808-0196, Japan; 15965052mt@gmail.com (T.M.); noriyasuhara@gmail.com (N.H.)

**Keywords:** endospore, dipicolinic acid, surface-enhanced Raman scattering, functional silver nanoparticles, terbium complex

## Abstract

In spore-forming bacteria such as *Bacillus* and *Clostridium*, the vegetative cells form highly durable hard shells called endospores inside the bacteria to survive as the growth environment deteriorates. Because of these properties, endospores can cause food poisoning and medical accidents if they contaminate food, medicine, or other products, and it is required for technology to detect the spores at the manufacturing site. In this study, we focused on the surface-enhanced Raman scattering (SERS) method for the sensitive detection of dipicolinic acid (DPA), a molecular marker of endospores. We constructed Fe_3_O_4_/Ag core–shell functional silver nanoparticles that specifically bind to DPA, and investigated a method for the qualitative detection of DPA by SERS and the quantitative detection of DPA by fluorescence method using a terbium complex formed on the surface. As a result, the concentration of the functional silver nanoparticles constructed could detect spore-derived DPA by fluorescence detection method, and SERS was several tens of nM. The functionalized nanoparticles can detect DPA quantitatively and qualitatively, and are expected to be applied to detection technology in the production of food and pharmaceuticals.

## 1. Introduction

Various disinfection/sterilization operations and microbiological tests are conducted in the food industry to prevent food poisoning and food spoilage/deformation due to bacterial contamination. As a general disinfection method, the use of alcohol (70–80% ethyl alcohol) is often used in the food sanitation field. Sterilization methods include high-pressure steam sterilization using an autoclave at 121 °C for 15–20 min at 2 atmospheres, dry heat sterilization using a dry heat sterilizer at 160 °C or higher for at least 30 min, boiling sterilization using boiling water, ultraviolet sterilization using ultraviolet radiation, and gas sterilization using ethylene oxide or propylene oxide [1,2]. In the pharmaceutical industry, raw materials are carefully cleaned and wiped from the time of delivery to minimize contamination by foreign substances and microorganisms. Sterilization methods include heat sterilization, moist heat sterilization, dry heat sterilization, irradiation sterilization, and sterilization by gas or fumigants, as described above. If heat sterilization is not possible for a liquid product, it is filtered and sterilized using a sterile filter of 0.22 μm or less or a filter to catch microorganisms. The quality and hygiene of all processes from shipping delivery are monitored.

General nutritional vegetative bacteria can be sterilized using these methods because of their weak resistance. However, some bacteria that form durable structures called spores are resistant to heat, drying, ultraviolet rays, and chemicals, and cannot be easily sterilized by ordinary sterilization methods [3,4,5]. Bacteria that form such endospores include *Bacillus* and *Clostridium* species. These bacteria form spores from the vegetative cells to survive when the surrounding nutritional status or growth environment deteriorates. When the nutritional status of the surrounding environment recovers, the bacteria germinate from the endospores, become nutrient cells, and repeatedly divide and multiply. When these spore-forming bacteria are mixed into food or products, they remain in the food as endospores even after the food has been sterilized, and the bacteria germinate and multiply from there, causing food poisoning and food spoilage or loss [6,7,8,9]. Therefore, contamination with spore-forming bacteria has become a problem in food and pharmaceutical manufacturing sites. In the current detection methods, samples are cultured and the colonies are counted, so it takes several days to a week to obtain the results. In addition, endospore detection methods targeting dipicolinic acid (DPA), a molecular marker of endospores, have been developed, including an absorption method based on the chelation of DPA with metal ions [10,11,12], a fluorescence method based on the adsorption of DPA with lanthanides [13,14,15,16], and a highly sensitive fluorescence method based on the complexation of terbium complexes with DPA [17,18,19,20]. However, the absorption method has low sensitivity, while the fluorescence method has very high sensitivity but has the disadvantage of being strongly affected by foreign substances. Against this background, there is a strong need for a technology to detect endospores in situ, rather than as bacteria, as a rapid and reliable detection method.

In this study, we developed a novel detection method using functionalized silver nanoparticles for simple and sensitive spore detection. To detect spores qualitatively with high sensitivity, we focused on surface-enhanced Raman scattering (SERS) [21,22,23,24]. The SERS effect occurs on metal surfaces such as gold, silver, and copper, and the enhancement effect is especially remarkable when silver is used [25,26,27]. In this study, we constructed functional silver nanoparticles that specifically bind to DPA. A terbium complex is formed on the surface layer of the functionalized silver nanoparticles, and when DPA is coordinated to this complex, it shows very high fluorescence, which enables highly sensitive and quantitative detection. Additionally, since the SERS effect is remarkably obtained in the gaps between the aggregated nanoparticles [24], we used Fe_3_O_4_/Ag core–shell nanoparticles with a magnetic Fe_3_O_4_ core for easy aggregation. In addition, the magnetic nature of the nanoparticles allows them to be separated and collected by magnets, thus removing impurities. Therefore, the Raman signal enhanced by SERS allows qualitative analysis of the DPA signal. This study aims to establish a simple, sensitive, quantitative, and qualitative detection technique for spores by using these functional silver nanoparticles.

## 2. Materials and Methods

### 2.1. Materials and Reagents

2,6-Pyridinedicarboxylic acid, iron(II) sulfate heptahydrate, potassium nitrate, sodium hydroxide, silver nitrate, trisodium citrate dehydrate, hydroxylammonium chloride, oxalic acid, ammonium oxalate monohydrate, and terbium (III) chloride hexahydrate were obtained from Wako Pure Chemical Industries (Japan). Branched polyethylenimine, Mw ~25,000, was obtained from Sigma Aldrich. Standard suspension of *Bacillus subtilis* spore (1.0 × 10^7^ CFU/mL, Eiken Chemical Co., Ltd., Tokyo, Japan) was used as a target endospore for extraction of DPA.

### 2.2. Extraction of DPA from Endospore

Extraction of DPA was performed by using an electroporation system (Gene Pulser, Bio-Rad, Hercules, CA, USA). The DC pulse voltage (1.6 kV/cm, 1.0 ms) was applied to the endospore standard solution. The details of this method were reported previously [28].

### 2.3. Synthesis of Functional Silver Nanoparticles

Fe_3_O_4_ nanoparticles were prepared in a previous report [29]. To improve the dispersibility, the PEI was prepared using a molecular weight of 25,000 g/mol. MilliQ water (80 mL) and FeSO_4_ (0.7 g) were added and mixed in a Nass flask. To this solution, KNO_3_ (10 mL, 2.0 M), NaOH (10 mL, 1.0 M), and PEI (0.4 g, 4 g/L) were added dropwise, and the mixture was heated and stirred at 90 °C for 2 h. Then, the solution was cooled at room temperature, and the Fe_3_O_4_ nanoparticles were trapped using a neodymium magnet and magnetically separated from the reaction mixture. The collected Fe_3_O_4_ nanoparticles were washed five times with MilliQ water and suspended in MilliQ water (80 mL) to obtain PEI-dispersed Fe_3_O_4_ nanoparticles (3.2 g/L, pH 7).

Silver nanoparticles were prepared as previously reported [30]. MilliQ water (97 mL) and AgNO_3_ solution (1 mL, 100 mM) were added to a beaker and heated until the solution started to boil. As soon as it started boiling, trisodium citrate solution (2 mL, 500 mM) was dropped into the AgNO_3_ solution. The heating was continued for another 15 min and then the solution was cooled at room temperature to obtain citric-acid-dispersed silver nanoparticles solution (Appendix A).

Fe_3_O_4_/Ag core–shell nanoparticles were prepared by the seeded particle growth method based on previous reports [29]. A suspension of Fe_3_O_4_ nanoparticles (2 mL) was added to a round-bottom flask and sonicated for 2 min in a tabletop ultrasonic cleaner (42 kHz). Then, the prepared silver nanoparticle solution (90 mL) was added and stirred for 2 h. Fe_3_O_4_–Ag seeds were magnetically separated from the excess silver nanoparticles solution, washed with MilliQ water five times, and suspended in MilliQ water (20 mL). PEI (0.04 g, 2 g/L) was added and heated in an oven at 60 °C for 1 h. After washing with MilliQ water 5 times, MilliQ water (20 mL) was added and Fe_3_O_4_–Ag seed-PEI was dispersed by sonication. Then, Ag shells were grown by iterative reduction of AgNO_3_ on Fe_3_O_4_–Ag seed-PEI by the following method. NaOH (110 mL, 0.01 M) was added to the Fe_3_O_4_–Ag seed-PEI dispersion solution and stirred. AgNO_3_ (0.5 mL, 1%) and HONH_3_Cl (0.75 mL, 0.2 M) were added dropwise and reacted for 5 min, followed by AgNO_3_ (0.5 mL, 1%) and HONH_3_Cl (0.25 mL, 0.2 M) added dropwise for 5 min. This 10-min reaction was repeated five times in total. After that, the reaction mixture was magnetically separated and washed five times with MilliQ water. MilliQ water (20 mL) was added and dispersed by sonication to obtain Fe_3_O_4_/Ag core–shell nanoparticles. To dissolve the unreacted Fe_3_O_4_ nanoparticles, oxalic acid solution (20 mL, 0.6 M) was prepared by mixing oxalic acid (0.756 g) and ammonium oxalate (0.979 g) in MilliQ water (20 mL). The prepared oxalic acid solution (20 mL) was added to the Fe_3_O_4_/Ag core–shell nanoparticle solution and stirred for 2 h. The Fe_3_O_4_/Ag core–shell nanoparticles were magnetically separated from the reaction solution, washed with MilliQ water five times, and MilliQ water (2 mL) was added. The preparation of Fe_3_O_4_/Ag core–shell nanoparticles was carried out under non-magnetic conditions by using a stirrer.

Functional silver nanoparticles were prepared by modifying the surface layer of magnetic silver nanoparticles with citric acid and terbium. The prepared Fe_3_O_4_/Ag core–shell nanoparticles (10 mg) were added to 0.1 M citrate buffer solution (1 mL, pH 7) and allowed to react for 2 h in a tube rotator. Then, they were magnetically separated from the excess citrate buffer solution and washed with MilliQ water five times. TbCl_3_ (1 mL, 0.1 mM) was then added and allowed to react for 2 h. The excess TbCl_3_ was removed by magnetic separation, the obtained sample was washed with MilliQ water five times, and MilliQ water (1 mL) was added to obtain functionalized silver nanoparticles.

### 2.4. Characterization of Nanoparticles

Absorption spectra of nanoparticles were taken by UV–VIS spectrophotometer (V-550DS, JASCO). Fluorescent measurements were performed using a multi-microplate reader (ARVO sx, Perkin Elmer, Waltham, MA, USA). Samples for fluorescence evaluation were prepared by adding 20 μL of DPA sample to 180 μL of 10 mg/mL functionalized silver nanoparticle sample. The sample (200 μL) was added to a black 96-well immunoplate (Thermo Fisher Scientific, Waltham, MA, USA), and the fluorescence was measured at an excitation wavelength of 280 nm and a fluorescence wavelength of 545 nm. The nanoparticles’ size was measured and estimated using dynamic light scattering (DLS) measurements (DelsaMax PRO, Beckman Coulter, Brea, CA, USA).

The morphology of synthesized nanoparticles was characterized with a transmission electron microscope (TEM) (JEM-3010, JEOL, Tokyo, Japan). Then, 10 μL of the sample was prepared to be dropped on (GPS-C10 STEM Cu100P, Okenshoji Co., Ltd., Tokyo, Japan), and dried in a vacuum desiccator for 24 h. The sample was observed by TEM at an acceleration voltage of 200 kV. 

The nanoparticle dispersion solution was magnetically separated from the solvent with a neodymium magnet and vacuum dried for 1 day with a desiccator. SEM (scanning electron microscope) and EDX (energy-dispersive X-ray spectrometry) images of the Fe_3_O_4_/Ag core–shell nanoparticles were observed by ultra-high resolution field emission scanning electron microscope (FE-SEM) (SU9000, HITACHI, Tokyo, Japan) at 20 kV accelerating voltage. The structural properties of the dried nanoparticle were analyzed by X-ray diffraction (XRD) (RINT-2000, Regaku, Tokyo, Japan).

Qualitative detection of DPA by Raman measurement was performed using a high-speed laser Raman microscope (RAMANtouch VIS-NIR-KUF, Nanophoton, Osaka, Japan). Functional silver nanoparticles (180 μL) were added to DPA (20 μL) to prepare a sample. A sample on a glass plate was added dropwise and vacuum dried with a desiccator. The dried sample was analyzed by Raman spectroscopy with an excitation laser wavelength of 533 nm, an exposure time of 10 s, and an integration frequency of 5 times, and a laser power of 5 mW.

## 3. Results and Discussion

### 3.1. The Sensing System in This Study

The flow of endospore detection developed in this study is shown in Figure 1. DPA is rapidly extracted from the spores by the electroporation method, and a complex of the extracted DPA and functional silver nanoparticles is formed. After separation by a magnet, DPA is quantified by fluorescence measurement, and its qualitative nature is measured by Raman spectroscopy.

### 3.2. Preparation and Characterization of Functionalized Nanoparticles

There have been reports on the SERS effect by silver nanoparticles [31,32], and it has been reported that SERS occurring between particles of 50–100 nm in size is most effective. Based on this information, we designed the size of functional silver nanoparticles and synthesized them. TEM images of the prepared Fe_3_O_4_ nanoparticles are shown in Figure 2A. From TEM observation, it was confirmed that the particles were cubic, with a diameter of 45 ± 5 nm. XRD analysis confirmed the diffraction peaks originating from Fe_3_O_4_ (Figure 2C). The prepared Fe_3_O_4_ nanoparticles could be completely separated from the solution by applying a magnetic field (Figure 2B).

TEM images of the prepared Fe_3_O_4_/Ag core–shell nanoparticles are shown in Figure 3A. From the TEM observation, it was confirmed that the cubic Fe_3_O_4_ nanoparticles disappeared and spherical nanoparticles of 80 ± 10 nm in diameter were constructed. Compared with the TEM image of Fe_3_O_4_ nanoparticles (Figure 2A), the shape of the Fe_3_O_4_ nanoparticles changed from cubic to spherical due to the silver coating, and the thickness of the silver shell was about 17.5 nm. In the separation test using a magnet, the constructed nanoparticles, as well as Fe_3_O_4_ nanoparticles, could be completely separated from the solution by applying a magnetic field, indicating that they encapsulate Fe_3_O_4_ nanoparticles (Figure 3B). The results of XRD analysis are shown in Figure 3C, where the diffraction peaks derived from Ag were confirmed by XRD. The results of EDX analysis are shown in Figure 3D. From the results of elemental mapping images and EDX spectra, the Fe elements in the core and Ag elements in the shell were confirmed. From these results, we considered that the Fe_3_O_4_/Ag core–shell nanoparticles were prepared for the biosensing of the endospore.

### 3.3. Molecular Modification and Fluorescence Detection Characterization

The terbium complexes and DPA coordination modified on the particle surface of Fe_3_O_4_/Ag core–shell nanoparticles were confirmed by fluorescence measurements. In Table 1, while no significant increase in fluorescence intensity was observed at DPA concentrations (1~5 nM), an increase in fluorescence of the functional silver nanoparticle–DPA complex was observed at each DPA concentration (10~1000 nM).

The change in the fluorescence intensity of the functional silver nanoparticle–DPA complex for each DPA concentration (10~1000 nM) prepared in the DPA standard solution is shown in Figure 4. The correlation coefficient of the DPA standard curve is 0.9949, indicating a very good linear relationship between the fluorescence intensity and the DPA concentration. In this standard curve, the detection limit (lower limit) of DPA with functionalized silver nanoparticles is 10 nM.

This excellent fluorescence property is attributed to the complexation of Tb and DPA of the functional silver nanoparticles, which results in a charge–transfer transition from the ligand DPA to the central metal Tb, resulting in increased fluorescence emission. The Tb–DPA complex is excited from the ground singlet state (S0) to the excited singlet state, which is then relaxed to the lowest excited singlet state (S1) by non-radiative transitions. The energy of the excited singlet state (S1) of DPA is then transferred to the excited triplet state (T1) via inter-term crossing (ISC). Subsequently, a charge transfer transition (CT) occurs from the excited triplet state (T1) of DPA to Tb (5D4), which emits a very strong fluorescence. In particular, the energy levels of Tb are better matched to those of DPA than those of other Ln (Dy, Gd, Eu, Sm), and the energy transfer occurs more efficiently [33]. Tb has fluorescence maxima at 490 nm, 544 nm, 590 nm, and 625 nm and belongs to the 5D4 → 7F6, 5D4 → 7F5, 5D4 → 7F4, and 5D4 → 7F3 transitions, respectively (Appendix A). The fluorescence emission at 544 nm is the strongest and is used as the fluorescence measurement wavelength. The enhancement of the fluorescence intensity due to this charge–transfer transition suggests that the modification of citric acid and Tb on the particle surface allows the sensitive fluorescence detection of DPA.

Using spore samples, the increase in fluorescence intensity of DPA extracts from spores by electroporation and a mixture of DPA extracts and functional silver nanoparticles germinated by reaction with L-Alanine for 120 min are shown in Figure 5. The DPA content in each sample was calculated from the increase in fluorescence intensity using the DPA standard curve determined above. The DPA concentration in the extracts treated by electroporation was about 80 nM, and that in the extracts treated with L-Alanie for 120 min was about 500 nM. This is close to the concentration of DPA quantified by liquid chromatography in a previous paper [28], suggesting that quantitative detection by fluorescence measurement is possible.

### 3.4. Qualitative Analysis of DPA by Functionalized Silver Nanoparticles and Raman Spectroscopy

First of all, the Raman measurement of DPA powder was carried out. The Raman spectrum of the DPA powder is shown in Appendix A. DPA consists of a pyridine ring with two carboxyl groups. The measured Raman shift of 400~1800 cm^−1^ contains many characteristic peaks that are useful for the identification of DPA. The Raman spectral bands of DPA powder and their attributions are shown in Appendix A and Appendix A [34]. The pyridine ring has a characteristic peak of a very strong symmetric ring “breathing” around 1000 cm^−1^. Other ring vibration modes include a C-C ring bend near 640 cm^−1^, trigonal ring “breathing” near 1080 cm^−1^, C-C ring mode near 1260 cm^−1^, C-C ring stretch near 1440 cm^−1^, and ring stretch near 1570 cm^−1^. C-H bends exist around 1150 cm^−1^, 1170 cm^−1^ and 1270 cm^−1^. C-H out-of-plane is present around 795 cm^−1^ and 885 cm^−1^, C-CO_2_ bend is present around 845 cm^−1^, and C-O stretch is present around 1315 cm^−1^. The carboxyl groups include OCO in-plane deformation at around 750 cm^−1^ and C=O stretch at 1640 cm^−1^.

The qualitative evaluation of the functionalized silver nanoparticle–DPA complex was carried out based on the peak information of the Raman spectral bands. The Raman spectra of trisodium citrate dihydrate and terbium (III) chloride hexahydrate used as functional molecules are shown in Figure 6A,B, respectively. Figure 6A shows several peaks originating from the citric acid structure. The peak from terbium chloride was observed around 1600 cm^−1^ (Figure 6B). The Raman spectra of the constructed functionalized silver nanoparticles are shown in Figure 6C. In the Raman spectra of the functionalized silver nanoparticles, no peaks derived from functional molecules were observed. The SERS effect is caused by the resonance effect due to the electron transfer interaction between the metal surface and the adsorbed molecules, and is more likely to occur with unsaturated hydrocarbons such as aromatic compounds. Since citric acid is a saturated hydrocarbon, the signal enhancement by SERS is weak, which is considered to be the reason why the peak could not be confirmed. The Raman spectra of the functionalized silver nanoparticles–DPA complexes complexed with DPA standard solution (1 μM) are shown in Figure 6D. The complexation of functionalized silver nanoparticles with DPA confirms the SERS effect: symmetric ring “breathing” around 1000 cm^−1^, C-O stretch around 1350 cm^−1^, and C-O stretch around 1440 cm^−1^. Symmetric ring “breathing” around 1000 cm^−1^, C-O stretch around 1350 cm^−1^, and C-C ring stretch around 1440 cm^−1^ were identified as DPA-derived peaks. In metallic nanoparticles such as gold, silver, and copper, SERS occurs, in which Raman scattered light from materials adsorbed on the surface is enhanced [31,32,35]. This phenomenon is thought to be due to the effect of the enhanced electric field of the localized surface plasmon resonance [35]. When light hits the nanoparticles, they are covered by a strong electric field due to the localized surface plasmon resonance. A very strong enhanced electric field is generated at the contact point of the particles covered by the strong electric field. Molecules at that contact point experience significant SERS effects. The enhanced electric field is surface selective because it decays exponentially with distance from the surface. The other reason it is thought to occur is resonance effects due to electron transfer interactions between the metal surface and the adsorbed molecules. In this case, the molecule must resonate against the transfer of electrons from the metal to the adsorbed molecule or from the adsorbed molecule to the metal. In particular, unsaturated hydrocarbons have a greater resonance effect than saturated hydrocarbons [36]. The reason for the SERS effect compared with Figure 6C was considered to be the resonance effect caused by the electron transfer interaction between the unsaturated hydrocarbon DPA and the metal, and the signal enhancement caused by the very strong electric field generated by the aggregation of nanoparticles.

The Raman spectra of the functional nanoparticles reacted with the solution extracted from the endospores by the electroporation method are shown in Figure 7. For the extracted solution, no Raman spectral peaks were observed (Figure 7a). On the other hand, the Raman spectra of the extract solution (10-fold dilution and no dilution) and the complexed functional silver nanoparticles are shown in Figure 7c,d, respectively. The comparison of these spectra clearly shows the enhancement of the Raman spectra, which confirms the SERS effect of functionalized silver nanoparticles. From the enhanced spectra, the symmetric ring “breathing” around 1001 cm^−1^, C-C ring mode around 1260 cm^−1^, C-O stretch at 1337 cm^−1^, and ring stretch at 1574 cm^−1^ can be seen. The combination of functionalized silver nanoparticles made it possible to identify multiple DPA-derived peaks in the extract from the spores, indicating the possibility of the qualitative analysis of DPA from complex Raman signals. However, the samples of diluted extract and functionalized silver nanoparticles did not show any increase in peaks that could be used for the qualitative evaluation of DPA. The SERS effect is greatly influenced by the adsorption (binding) state on the metal surface, the shape of the surface, and the aggregation state. Therefore, to further improve the sensitivity of the qualitative analysis by SERS, it is necessary to optimize the size and shape of the particles, the functional molecules that adsorb DPA, and the aggregation state of the particles after the reaction.

## 4. Conclusions

The sensor developed in this study is a simple, rapid, and accurate detection technology for such bacterial spores, and we believe that it is an essential technology for the realization of a safe and secure society, including food and pharmaceutical manufacturing sites. In addition, anthrax, which has been a hot topic for a while, forms similar spores, so this technology can also contribute to countermeasures against bioterrorism. In general, it takes three days to detect endospores from a food sample [37], and several hours even for a method that identifies the spore-forming bacteria by PCR [38]. Rapidity is especially important for detection that requires urgency. In this study, we succeeded in constructing functional Ag nanoparticles detectable by fluorescence and SERS methods for the highly sensitive detection of endospores with rapidity, and the concentration at which spore-derived DPA could be quantitatively and qualitatively detected was several tens of nM. However, the concentration of spore-derived DPA required in the food industry is a few nM. In addition, the extraction and reliable detection of DPA from spores in food is important. The SERS effect is greatly affected by the state of adsorption on the metal surface, the shape of the surface, and the aggregation state, which in turn affects the signal enhancement. In order to further increase the sensitivity of this technology for practical use, it is necessary to design particles and functional molecules and to develop methods to control the dispersion and aggregation of the particles. Additionally, when combined with a method that can extract DPA from spores in food, it will be a very powerful spore-detection tool.

## Figures and Tables

**Figure 1 biosensors-12-00068-f001:**
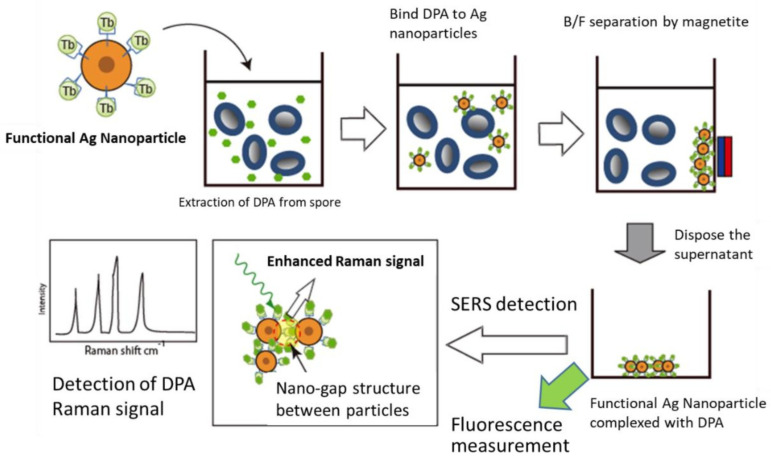
Schematic diagram of endospore detection system using functional silver nanoparticles.

**Figure 2 biosensors-12-00068-f002:**
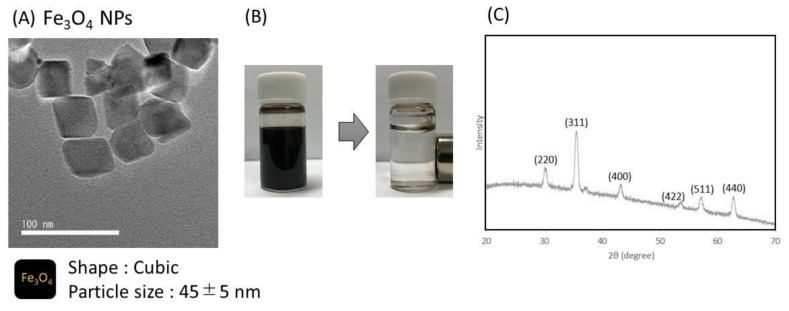
Characterization of Fe_3_O_4_ nanoparticles. (**A**) TEM image, (**B**) image of separation by magnet, and (**C**) XRD analysis of Fe_3_O_4_ nanoparticles.

**Figure 3 biosensors-12-00068-f003:**
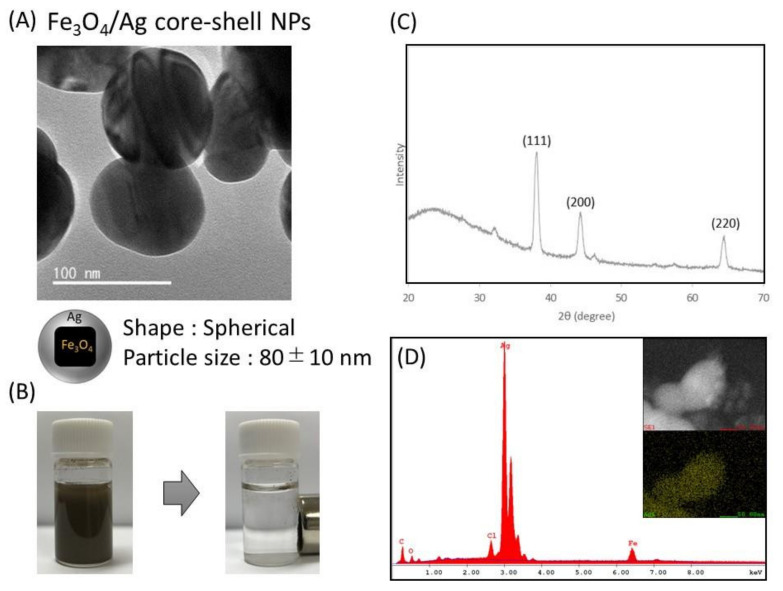
Characterization of Fe_3_O_4_/Ag core–shell nanoparticles. (**A**) TEM image, (**B**) image of separation by magnet, (**C**) XRD analysis of core–shell nanoparticles, and (**D**) SEM-EDX analysis of core–shell nanoparticles. The above image is the SEM image and the following yellow is the Ag mapping image.

**Figure 4 biosensors-12-00068-f004:**
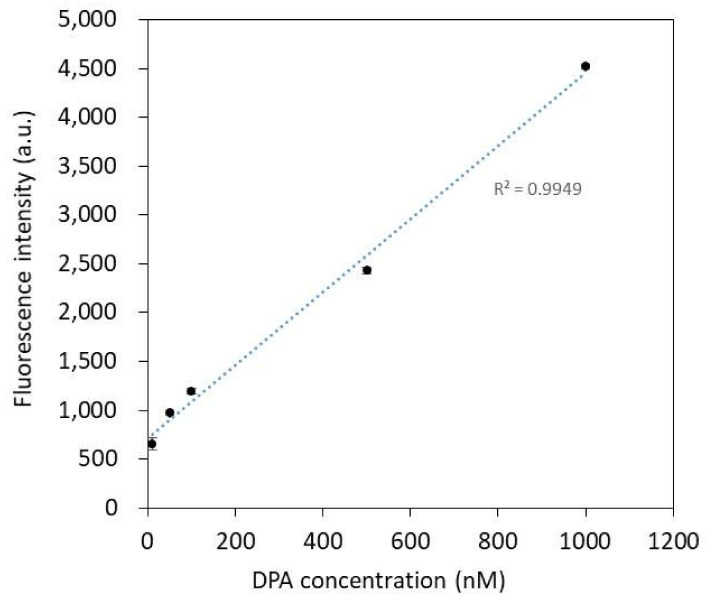
Calibration curve for DPA concentrations from 0 to 1 μM monitored at 545 nm in a microplate fluorometer. The correlation coefficient of the DPA standard curve is 0.9949. Error bars represent means ± SD (*n* = 5).

**Figure 5 biosensors-12-00068-f005:**
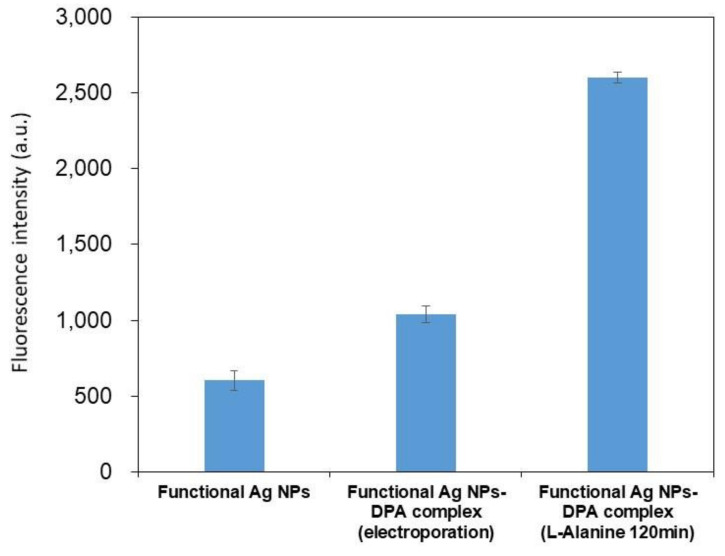
Increase in fluorescence intensity of DPA extract samples from spores by electroporation method, and germinated samples by reaction with L-alanine for 120 min mixed with functional silver nanoparticles. Error bars represent means ± SD (*n* = 5).

**Figure 6 biosensors-12-00068-f006:**
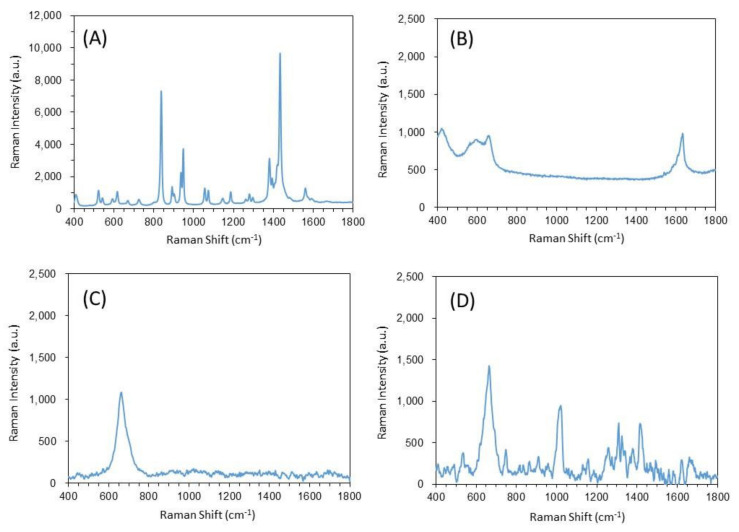
The qualitative evaluation by Raman spectra. (**A**) Trisodium citrate dihydrate, (**B**) terbium (III) chloride hexahydrate, (**C**) functionalized silver nanoparticle, and (**D**) functionalized silver nanoparticle-DPA complex.

**Figure 7 biosensors-12-00068-f007:**
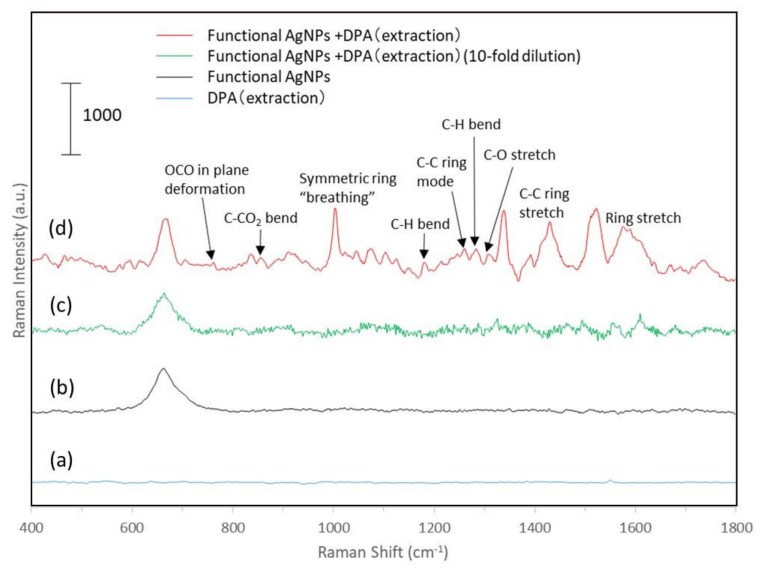
Raman spectra of the reaction of solution extracted from spores by electroporation method with functional Ag nanoparticles. (**a**) DPA extraction sample, (**b**) functional Ag nanoparticles, (**c**) 10-fold dilution of DPA extraction sample with functional Ag nanoparticles, and (**d**) DPA extraction sample with functional Ag nanoparticles.

**Table 1 biosensors-12-00068-t001:** Fluorescence intensity of the functional silver nanoparticle–DPA complex.

DPA Concentration (nM)	Fluorescence Intensity (a.u.)
0	604
1	603
5	608
10	662
50	974
100	1194
500	2433
1000	4518

The data represent the mean of five samples each, and a two-sample *t*-test assuming equal variance indicates a statistically significant difference at concentrations above 10 nM at the 5% significance level.

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
