# Peer review of "Multi-Functional Silver Nanoparticles for High-Throughput Endospore Sensing"

_biosensors, 2022, doi:10.3390/bios12020068_

Round 1

Reviewer 1 Report

The authors presented the novel method using functionalized silver nanoparticles for simple and sensitive spore detection. The development of a fast, efficient and simple method for the detection of spores at the manufacturing site is important because endospores can cause food poisoning and medical accidents if they get into food, medicine, or other products. In the presented study the authors constructed functional silver nanoparticles that specifically bind to DPA. A terbium complex is formed on the surface layer of the functionalized silver nanoparticles, and when DPA is coordinated to this complex, it shows very high fluorescence, which enables highly sensitive and quantitative detection. Moreover, the magnetic nature of the nanoparticles allows them to be separated and collected by magnets, thus removing impurities. The authors used Surface-enhanced Raman Scattering (SERS) to detect spores qualitatively with high sensitivity. 

The experiments are well planned and carried out carefully step by step. The paper is well written.

The paper may be considered to publish after supplementing according to the following questions and comments: 

  1. The authors wrote “The change in the fluorescence intensity of the functional silver nanoparticle-DPA complex for each DPA concentration (10~1000 nM) prepared in the DPA standard solution is shown in Figure 4. The correlation coefficient of the DPA standard curve is 0.9949, indicating a very good linear relationship between the fluorescence intensity and the DPA concentration. In this standard curve, the detection limit (lower limit) of DPA with functionalized silver nanoparticles was 10 nM.”
  • Question1: The authors analyzed the dependence of fluorescence intensity of complexed DPA by DPA concentration, which enabled the preparation of a calibration curve and determining the detection limit. But there is no information about the yield of complex formation. Do the authors analyze the non-complex DPA by another analytical method to determine the yield of complex formation?
  • Question 2: Did the authors determine the loading of Fe3O4/Ag core-shell nanoparticles to calculate the amount of material (silver nanoparticle) for complex formation. How much (mg?) did you use Fe3O4/Ag core-shell nanoparticles for complexing the 5 mM (or 50 mM) of PDA?
  1. Authors wrote: “The excess TbCl3 was magnetically separated from the excess TbCl3, washed with MilliQ water five times, and MilliQ water (1 mL) was added to obtain functionalized silver nanoparticles.”
  • Comment: There is a repetition in the sentence. Please correct.

  1. Question: Are the author going to test their method on food samples? How will the preparation of the sample look like then?

Author Response

In general, we appreciate the constructive comments and suggestions made by reviewers. We have tried to incorporate all into a revised manuscript. Please find below, the point-by-point changes that we introduced.

Reviewer comment:
Question1: The authors analyzed the dependence of fluorescence intensity of complexed DPA by DPA concentration, which enabled the preparation of a calibration curve and determining the detection limit. But there is no information about the yield of complex formation. Do the authors analyze the non-complex DPA by another analytical method to determine the yield of complex formation?

Author response:
We are very grateful for your suggestion. In this study, we found that the terbium-citrate complex is adsorbed on the surface layer of silver nanoparticles and shows very strong fluorescence when DPA is coordinated. At the same time, a small amount of citrate-Tb-DPA can be released, making it difficult to accurately determine the non-complexed DPA. Therefore, in this study, after mixing the samples, we did not separate the unbound samples and detected them directly. As shown in the reference papers17,18, to investigate the usefulness of AgNP-Tb3+ as a fluorescent sensor for DPA detection, a series of DPA solutions with concentrations ranging from 0 to 1 μM were evaluated by adding each solution to a fixed concentration of AgNP-Tb3+ solution. The title of the Figure 5 was not appropriate and has been corrected.

Reviewer comment:
Question 2: Did the authors determine the loading of Fe3O4/Ag core-shell nanoparticles to calculate the amount of material (silver nanoparticle) for complex formation. How much (mg?) did you use Fe3O4/Ag core-shell nanoparticles for complexing the 5 mM (or 50 mM) of PDA?

Author response:
Thank you for your suggestion. First of all, there was a lack of description about the volume and size of functional Ag nanoparticles, so we have added the following information.

L179-182
“There have been reports on the SERS effect by silver nanoparticles [31,32], and it has been reported that SERS occurring between particles of 50-100 nm in size is most effective. Based on this information, we designed the size of functional silver nanoparticles and synthesized them.”

In this experiment, 20uL of DPA sample was added to 180uL of 10mg/mL functional silver nanoparticle sample for fluorescence evaluation. This point has also been added in section 2.4.

L143-145
“Samples for fluorescence evaluation were prepared by adding 20 μL of DPA sample to 180 μL of 10 mg/mL functionalized silver nanoparticle sample.”

Reviewer comment:
Authors wrote: “The excess TbCl3 was magnetically separated from the excess TbCl3, washed with MilliQ water five times, and MilliQ water (1 mL) was added to obtain functionalized silver nanoparticles.”

  • Comment: There is a repetition in the sentence. Please correct.

Author response: Thank you for your feedback and point out this. We have corrected this mistake. The sentence has been revised as follows.

L137-138
“The excess TbCl3 was removed by magnetic separation, the obtained sample was washed with MilliQ water five times, and MilliQ water (1 mL) was added to obtain functionalized silver nanoparticles.”

Reviewer comment:
Question: Are the author going to test their method on food samples? How will the preparation of the sample look like then?

Author response: Thank you very much for your valuable question. In the future, we aim to evaluate it on food samples. With this sensor, qualitative evaluation can be expected by Raman signals, but for this purpose, the pretreatment method is important to detect spores in the sample, as shown in this reference paper [37]. In this paper, spores were separated from other microorganisms in the sample by colony counting after heat treatment of sample. If the technology to separate spores in food samples can be developed, this sensor will be of great value. In the future, we aim to develop a technology that can reliably detect DPA components even in the presence of contaminants.

37.Vandeweyer, D., Lievens, B. and Van Campenhout, L. Identification of bacterial endospores and targeted detection of foodborne viruses in industrially reared insects for food. Nat. Food, 2020; 1, 511–516.

Reviewer 2 Report

The manuscript ID biosensors-1559383 mainly presents a particular study about a surface-enhanced Raman scattering method assisted by silver nanoparticles for detecting dipicolinic acid with potential applications as a molecular marker of endospores. A list of comments for the authors is below:

  1. It is not clear how was selected the volume fraction of the different elements integrating the Fe3O4/Ag core-shell nanoparticles for this work.
  2. Please comment about the reproducibility and statistics in the measurements reported.
  3. Precipitation or agglomeration of the nanoparticles can be observed in the samples studied during measurements?
  4. The authors state that “Absorption spectra of nanoparticles were taken by UV–VIS spectrophotometer (V-137 550DS, JASCO).” But the data are missing in the manuscript.
  5. Please comment about the time of degradation of the functionalized Ag nanoparticles employed in this study to be considered for real applications.
  6. Please describe the facts related to the error bar in the fluorescence measurements. Besides, the error bar in Raman evaluations must be reported.
  7. A deeper description of the plasmonic assistance exhibited by Ag nanoparticles in the biosensing performance must be described. Modification in potential plasmonic coupling interactions can emerge from the dependence of the Raman observations on irradiance and wavelength selected for the optical measurements. You can see for instance: https://doi.org/10.1364/AO.383156
  8. Advantages and disadvantages of this technique must be confronted with updated publications in the same topic. You can see for instance: https://doi.org/10.1038/s43016-020-0120-z
  9. The collective citations presented in this work ought to be split in order to justify with individual expressions the importance of each reference to be included in the bibliography. Moreover, several references could be updated.
  10. The resolution of the system as a sensing platform is sensitive to light? Or the experiments must be made in darkness?

Author Response

In general, we appreciate the constructive comments and suggestions made by reviewers. We have tried to incorporate all into a revised manuscript. Please find below, the point-by-point changes that we introduced.

Reviewer comment 1:
It is not clear how was selected the volume fraction of the different elements integrating the Fe3O4/Ag core-shell nanoparticles for this work.

Author response:

Thank you reviewer for your comment and to point out this. There are several reports on the SERS effect of silver nanoparticles [31,32], and it is reported that the most effective is between particles of 50 to 100 nm. Based on this information, magnetic core particles are prepared and coated with a silver shell. The size of the obtained functional silver nanoparticles is about 80 nm, which is the expected particle size. This discussion has been added to the Results and Discussion section.

L179-182
“There have been reports on the SERS effect by silver nanoparticles [31,32], and it has been reported that SERS occurring between particles of 50-100 nm in size is most effective. Based on this information, we designed the size of functional silver nanoparticles and synthesized them.”

31.Guo H., Zhang Z., Xing B., Mukherjee A., Musante C., White J.C., and He L.
Analysis of Silver Nanoparticles in Antimicrobial Products Using Surface-Enhanced Raman Spectroscopy (SERS)
Environmental Science & Technology, 2015; 49 (7), 4317-4324.

32. Stamplecoskie K.G., Scaiano J.C., Tiwari V.S., and Anis H.,
Optimal Size of Silver Nanoparticles for Surface-Enhanced Raman Spectroscopy
The Journal of Physical Chemistry C 2011 115 (5), 1403-1409

Reviewer comment 2:
Please comment about the reproducibility and statistics in the measurements reported.

Author response:
Thank you for your suggestion. For fluorescence analysis, the data represent the mean of five samples each, and a two-sample t-test assuming equal variances indicated a statistically significant difference at concentrations above 10 nM at the 5% significance level. This is mention in Table 1. As for reproducibility, we believe that DPA can be detected quantitatively from the error bars shown in Figure 4.

Reviewer comment 3:
Precipitation or agglomeration of the nanoparticles can be observed in the samples studied during measurements?

Author response:
Thank you for providing these insights. There is no precipitation or agglomeration when measuring fluorescence. When we evaluate Raman spectra, we evaluate them by agglutination.

Reviewer comment 4:
The authors state that “Absorption spectra of nanoparticles were taken by UV–VIS spectrophotometer (V-137 550DS, JASCO).” But the data are missing in the manuscript.

Author response:
Thank you reviewer to point out this. During the preparation of functional silver nanoparticles, the particle size distribution and absorption spectrum of silver nanoparticles were checked. The data has been added as new Figure S1.

Reviewer comment 5:
Please comment about the time of degradation of the functionalized Ag nanoparticles employed in this study to be considered for real applications.

Author response:
Thank you for your comments, and point out this. The degradation of these functional nanoparticles can be attributed to the desorption of the functional molecules (citric acid-terbium complexes) adsorbed on the surface, causing the particles to aggregate together. We have not calculated the specific shelf life of the product, but as far as we have used it in the laboratory, we have stored it in the refrigerator for several months without any problems. We believe that it is possible to store the product for longer than that, but we have not yet evaluated the limits. We believe that the product is sufficient for actual use, but further studies to improve the shelf life will be necessary in the future.

Reviewer comment 6:
Please describe the facts related to the error bar in the fluorescence measurements. Besides, the error bar in Raman evaluations must be reported.

Author response: Thank you for pointing out. We have added information about error bars in Figure 5. As for Raman spectra, we believe that it is difficult to obtain quantitative data because the enhancement of Raman signals varies depending on the degree of aggregation between particles and the measurement location. Therefore, only qualitative data is presented by analyzing the peaks of the spectrum.

Reviewer comment 7:
A deeper description of the plasmonic assistance exhibited by Ag nanoparticles in the biosensing performance must be described. Modification in potential plasmonic coupling interactions can emerge from the dependence of the Raman observations on irradiance and wavelength selected for the optical measurements. You can see for instance: https://doi.org/10.1364/AO.383156

Author response: Thank you for your valuable comments. We have discussed the augmentation effect of Ag nanoparticles in this sensing and added the following to the text. The revised and added parts are shown in red.

L288-301
“In metallic nanoparticles such as gold, silver, and copper, SERS occurs, in which Raman scattered light from materials adsorbed on the surface is enhanced [31,32,35]. This phenomenon is thought to be due to the effect of the enhanced electric field of the localized surface plasmon resonance [35]. When light hits the nanoparticles, they are covered by a strong electric field due to the localized surface plasmon resonance. A very strong enhanced electric field is generated at the contact point of the particles covered by the strong electric field. Molecules at that contact point experience significant SERS effects. The enhanced electric field is surface selective because it decays exponentially with distance from the surface. The other reason is thought to occur is from resonance effects due to electron transfer interactions between the metal surface and the adsorbed molecules. In this case, the molecule must resonate against the transfer of electrons from the metal to the adsorbed molecule or from the adsorbed molecule to the metal. In particular, unsaturated hydrocarbons have a greater resonance effect than saturated hydrocarbons [36].

35. Futamata M., Yu Y., Yajima T., Elucidation of electrostatic interaction between cationic dyes and Ag nanoparticles generating enormous SERS enhancement in aqueous solution, J. Phys. Chem. C, 2011; 115, 5271-5279.

36. Otto A., The ‘chemical’ (electronic) contribution to surface-enhanced Raman scattering, J. Raman Spectrosc., 2005; 36, 497-509.

Reviewer comment 8:
Advantages and disadvantages of this technique must be confronted with updated publications in the same topic. You can see for instance: https://doi.org/10.1038/s43016-020-0120-z

Author response:
Thank you for your suggestion. We have revised our outlook by discussing the advantages and disadvantages of this study based on your suggested paper. The conclusions have been revised in red.

L338-353
In general, it takes three days to detect endospores from a food sample [37], and several hours even for a method that identifies the spore-forming bacteria by PCR [38]. Rapidity is especially important for detection that requires urgency. In this study, we succeeded in constructing functional Ag nanoparticles detectable by fluorescence and SERS methods for highly sensitive detection of endospores with rapidly, and the concentration at which spore-derived DPA could be quantitatively and qualitatively detected was several tens of nM. However, the concentration of spore-derived DPA required in the food industry is a few nM. In addition, the extraction and reliable detection of DPA from spores in food will be important. The SERS effect is greatly affected by the state of adsorption on the metal surface, the shape of the surface, and the aggregation state, which in turn affects the signal enhancement. In order to further increase the sensitivity of this technology for practical use, it is necessary to design particles and functional molecules and to develop methods to control the dispersion and aggregation of the particles. And when combined with a method that can extract DPA from spores in food, it will be a very powerful spore detection tool.”

37.Vandeweyer, D., Lievens, B. and Van Campenhout, L. Identification of bacterial endospores and targeted detection of foodborne viruses in industrially reared insects for food. Nat. Food, 2020; 1, 511–516.

38.Kędrak-Jabłońska A., Budniak S., Szczawińska A., Reksa M., Krupa M., and Szulowski K., Evaluation of real-time PCR based on SYBR Green I fluorescent dye for detection of Bacillus anthracis strains in biological samples, J. Vet. Res., 2018; 62(4), 549–554.

Reviewer comment 9:
The collective citations presented in this work ought to be split in order to justify with individual expressions the importance of each reference to be included in the bibliography. Moreover, several references could be updated.

Author response: Thank you for pointing out. The cited references that you pointed out have been divided and re-written for each content as fellow;

L54-61
In addition, endospore detection methods targeting dipicolinic acid (DPA), a molecular marker of endospores, have been developed, including an absorption method based on chelation of DPA with metal ions [10-12], a fluorescence method based on adsorption of DPA with lanthanides [13-16], and a highly sensitive fluorescence method based on complexation of terbium complexes with DPA [17-20]. However, the absorption method has low sensitivity, while the fluorescence method has very high sensitivity but has the dis-advantage of being strongly affected by foreign substances.

Reviewer comment 10:
The resolution of the system as a sensing platform is sensitive to light? Or the experiments must be made in darkness?

Author response:
Thank you for your feedback. Raman spectroscopy and fluorescence measurements are sensitive to light from outside, so they must be performed in the dark.

Round 2

Reviewer 1 Report

The manuscript was corrected according to the Reviewers' suggestion and may be considered to publish.

Reviewer 2 Report

Most of the points have been addressed in the reviewed version of the manuscript. Then, in my opinion, it can be considered for publication.